# First-Principle Prediction of Stress-Tunable Single-Photon Emitters at Telecommunication Band from Point Defects in GaN

**Junxiao Yuan** [1,†]**, Ke Wang** [1,†]**, Yidong Hou** [1]**, Feiliang Chen** [2,*]** and Qian Li** [3,*]

1 Department of Physics, Sichuan University, Chengdu 610065, China; yuanjunxiao@stu.scu.edu.cn (J.Y.); ke.wang@scu.edu.cn (K.W.); houyd@scu.edu.cn (Y.H.)
2 School of Electronic Science and Engineering, University of Electronic Science and Technology of China, Chengdu 610054, China
3 Microsystem and Terahertz Research Center, China Academy of Engineering Physics, Chengdu 610299, China
* Correspondence: flchen@uestc.edu.cn (F.C.); liqian_mtrc@caep.cn (Q.L.)
† These authors contributed equally to this work.

**Abstract:** Point defect-based single-photon emitters (SPEs) in GaN have aroused a great deal of interest due to their room-temperature operation, narrow line width and high emission rate. The room-temperature SPEs at the telecommunication bands have also been realized recently by localized defects in GaN in experiments, which are highly desired for the practical applications of SPEs in quantum communication with fiber compatibility. However, the origin and underlying mechanism of the SPEs remain unclear to date. Herein, our first-principle calculations predict and identify an intrinsic point defect $N_{Ga}$ in GaN that owns a zero-phonon line (ZPL) at telecommunication windows. By tuning the triaxial compressive strain of the crystal structure, the ZPL of $N_{Ga}$ can be modulated from 0.849 eV to 0.984 eV, covering the fiber telecommunication windows from the O band to the E band. Besides the ZPL, the formation energy, band structure, transition process and lifetime of the SPEs under different strains are investigated systematically. Our work gives insight into the emission mechanism of the defect SPEs in GaN and also provides effective guidance for achieving wavelength-tunable SPEs working in fiber telecommunication windows.

**Keywords:** single-photon emitters; atom defect; first-principle calculations; telecommunication band; stress

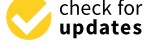



## 1. Introduction

As a central building block for quantum information technologies, single-photon emitters (SPEs) could be widely used in quantum secure communication, optical quantum computing, and so on [1–3]. An isolated two-level quantum system composed of an excited state and a ground state is necessary for an ideal SPE, where the electron only prefers to transit in this two-level quantum system by absorbing or emitting one photon at a time [4]. The early-stage SPEs are based on single-atom [5], however, the low reliability and efficiency severely limit their application. Quantum dots [6] and color centers [7] based on solid materials such as diamond [8] and transition metal dichalcogneides [9,10] are also alternative systems for producing SPE and showed an advantage in combination with advanced technologies of the semiconductor industry. In recent years, point defects in wide band gap semiconductors have been widely demonstrated as one of the best choices to realize room temperature (RT) SPEs [11]. Electron wave functions of defect atoms contribute to the formation of defect levels which could make up an isolated and heat-stabilized two-level system. However, the practical applications of SPEs on quantum communication require data transmission over optical fibers, and most of the RTSPEs are located in visible wavelengths so far, where the attenuation of photons in fiber is more than 3 dB/km and unsuitable for long-distance transmission [12]. On the other hand, the fiber telecommunication windows (from 1260 nm to 1675 nm, or from O to U band)

at near-infrared (NIR) are widely used in fiber-based long-distance telecommunications, benefiting from the low attenuation (0.3 dB/km for 1310 nm and 0.15 dB/km for 1550 nm). In the meanwhile, the solar radiation and Rayleigh scattering in the NIR fiber windows are also relatively low, and free-space telecommunication is also applicable in this waveband. Therefore, RT SPEs working in fiber telecommunication windows need to be exploited.

GaN is the most commercialized III-nitride material and has been developed to realize SPEs based on the localized point defects inside [13,14]. Its wide direct band gap [15] can suppress the interaction between the defect levels in the band gap and the bulk states, while the weak photoacoustic coupling could produce a zero-phonon line (ZPL) with narrow line width. The RT defect SPEs in GaN were first realized in an experiment by Berhane et al. in 2017 and the measured ZPL varied from 620 nm to 780 nm [16]. Then in 2018, high-performance RT SPEs in the telecom range (about 1000 nm–1300 nm) were further found in GaN [17], but their origin and defect atom structure are still unclear. As for theory, although the calculations of Zang et al. indicated that neutral $V_{Ga}$ can work as a potential SPE working in NIR and its ZPL can be tuned by the strain [18], the requirement of a specific p-type GaN to maintain its neutral state does not agree well with the experiment results. Therefore, much more effort should be paid to exploring the possible origin and emission mechanism of the defect SPEs in GaN operating in the fiber telecommunication band.

On the other hand, the influence of strain on SPEs has attracted wide interest recently, which can effectively modify emission properties such as wavelength and polarization. Gabriele's work has confirmed the non-monotonic behavior of the ZPL energy in hexagonal boron nitride under the effect of strain, and its role in the large spectral distribution was also observed experimentally [19]. This modulation effect of strain on ZPL wavelength provides a possible way to explain the emission in the fiber telecommunication band for SPEs in GaN and also inspires new applications, such as multidimensional information coding.

In this work, we carried out a systematic theoretical analysis of the intrinsic atom defects in GaN through the first-principle calculation and found a point defect $N_{Ga}$ that can produce SPE with ZPL in the fiber telecommunication band. By tuning the triaxial compressive strain of the crystal structure, the ZPL could be adjusted from the O band to the E band (1260 nm to 1460 nm). The formation energy, band structure, charge density, transition mechanism, and orbital composition of the point defect $N_{Ga}$ in GaN have been systematically investigated.

## 2. Calculation Methods

Our first-principle calculations based on the density functional theory (DFT) were performed by the PWmat package [20,21]. The atomic positions of the defect structure were optimized with the Perdew-Burke-Ernzerhof (PBE) functional [22], and the self-consistent field (SCF) calculations to give the band structure were performed with the Heyd-Scuseria-Ernzerhof (HSE) hybrid functional [23]. All calculations were spin-polarized, and the energy cutoff for the basis function was set to 70 Ry (Rydberg). Since strain calculation involving cell size change required higher calculating accuracy than the formation calculation, the k-point set mesh parameter in the strain relaxation calculation was set to $3 \times 3 \times 3$ while the mesh parameter in the formation calculation was set to $3 \times 3 \times 2$.

The bulk GaN has a typical wurtzite structure with space group P63mc. The defect calculations are carried out with a supercell consisting of $3 \times 3 \times 2$ primitive cells, as shown in Figure 1. The strain $\sigma$ is defined as $\sigma = a/a_0$, where $a_0$ is the optimal lattice constant without strain, and $a$ is the corresponding lattice constant with strains. The triaxial stain $\sigma < 1$ represents the compressive strain, while $\sigma > 1$ represents the tensile strain. The bulk GaN applies triaxial tensile ($\sigma = 100.5\%$) and compressive (99.5%) strains along the x-axis, y-axis, and z-axis directions, respectively. The lattice constant parameters corresponding to the supercell with $\sigma = 1$ are a = b = 9.504 Å, c = 10.323 Å, $\alpha = \beta = 90°$, and $\gamma = 120°$. The defect model $N_{Ga}$ ($Ga_N$) used in this article is the GaN supercell in which a Ga (or N) atom is replaced by an N (or Ga) atom.

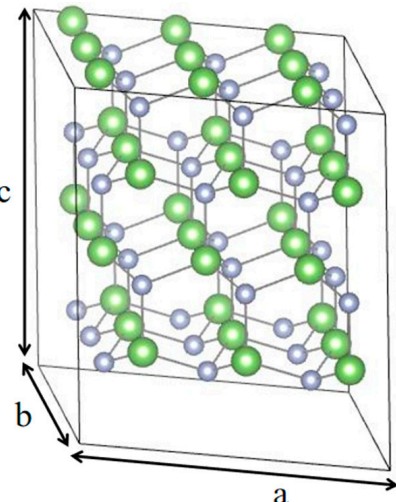

**Figure 1.** Supercell structure of the GaN crystal.

### 3. Results

*3.1. Formation Energy and Thermodynamic Stability of Single-Atom Defects*

  In our previous work [24], we demonstrated that defects $V_N$ and $N_i$ can be easily formed in experiments due to their low formation energy. However, their high concentration leads to the formation of a luminescence band rather than a stable single photon emitter (SPE), as the defects become too close to each other and become indistinguishable in space. In GaN, $V_N$ has been confirmed to cause a green luminescence band [25], while $N_i$ causes a yellow luminescence band [26]. On the other hand, $Ga_i$ has a high formation energy in N-rich conditions, which leads to a small concentration. However, its low migration energy reported in Ref. [27] suggests that it cannot serve as a stable SPE as it easily diffuses in GaN. Neutral $V_{Ga}$, with a high formation energy in both N-rich (7.9 eV) and Ga-rich (6.7 eV) conditions, has the potential to serve as an SPE, as suggested by the calculation results from Zang et al. [18]. However, its 5-electron-8-orbital electronic configuration is only stable in the lower part of the band gap according to the formation energy diagram. This indicates that a specific p-type GaN is required to maintain its neutral state, which is a challenge to be realized in an experiment. In terms of intrinsic point defects, $Ga_N$ and $N_{Ga}$ are the only possible defects that can serve as SPEs in telecommunication windows. Therefore, we focus our discussion on $Ga_N$ and $N_{Ga}$. Their thermodynamic stability is assessed by calculating their formation energies, which are as follows:

$$E_f(\alpha,q,E_F) = E_{tot}(\alpha,q) - E_{tot}(host) + \sum n_i \mu_i + q(E_F + E_{VBM}(host)) + E_{corr} \qquad (1)$$

where $E_{tot}(\alpha, q)$ is the total energy of the supercell with defect $\alpha$ in charge state q, and $E_{tot}(host)$ is the total energy of the same supercell without defect. $\sum N_i \mu_i$ is the chemical potential of atom i, and $n_i$ is the number of the atom *i* that removed from the cell. $E_{VBM}(host)$ is the energy eigenvalue of the valence band maximum (VBM) level which is aligned referenced to the electrostatic potential far from the defect site in the supercell. $E_F$ is the Fermi level referring to the VBM level. A high Fermi level usually requires a high electron concentration, i.e., an n-type environment. $E_{corr}$ is a corrected value caused by the long-range image-charge Coulomb interaction [28].

  Formula (1) shows that the line slope for formation energy $E_f(\alpha,q,E_F)$ is charge q. As seen in Figure 2, it is evident that among their own charged states, the neutral $N_{Ga}$ and neutral $Ga_N$ have the highest formation energy among their own charged states. The neutral $N_{Ga}$ can remain stable at a Fermi energy ranging from 1.7 eV to 2.6 eV (marked in red shadow), which is corresponding to a weak n-type doping environment for unintentional doping in GaN. On the other hand, $Ga_N$ requires a stronger n-type doping environment than $N_{Ga}$ and can remain stable at a Fermi energy ranging from 2.3 eV to 3.0 eV (marked in

gray shadow). The $N_{Ga}^{1.005}$ and $N_{Ga}^{0.995}$ are the 0 state $N_{Ga}$ with $\sigma = 100.5\%$ and $\sigma = 99.5\%$, respectively. $N_{Ga}^{0.995}$ is shifted up by 0.15 eV compared to $\sigma = 100\%$, while $N_{Ga}^{1.005}$ is shifted down by 0.03 eV. Despite these minor shifts, it is evident that the formation energy and stability of these defects would not be significantly affected by these strains.

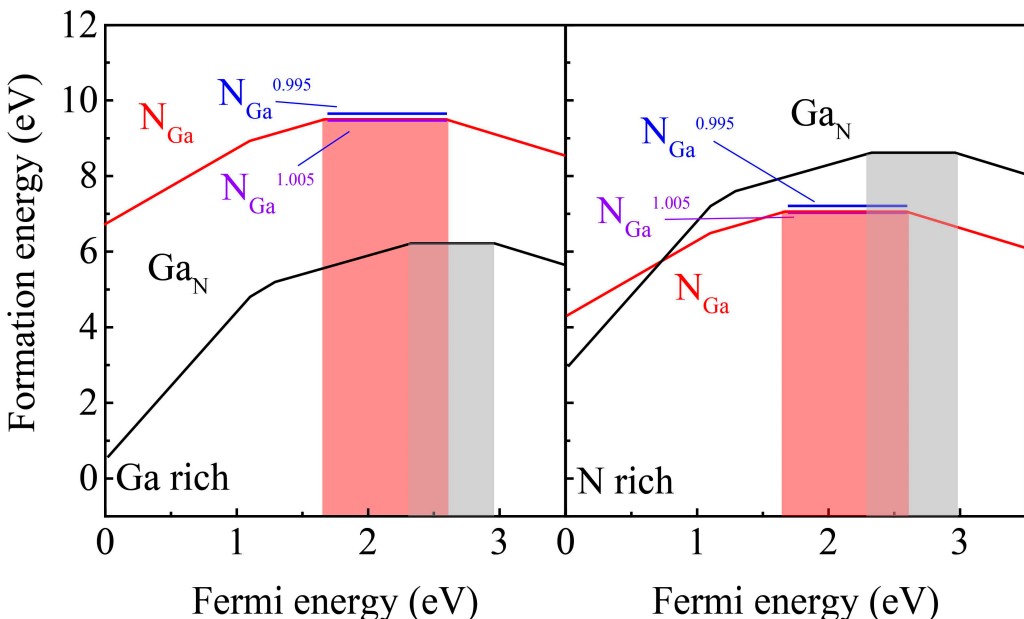

**Figure 2.** Formation energy of $Ga_N$ and $N_{Ga}$, as a function of the Fermi level in N rich and Ga rich GaN. The zero and maximum of the Fermi-level scale correspond to the top of the valence band and the bottom of the conduction band, respectively. The solid lines correspond to the formation energies for the most stable charge states of the defects. The $Ga_N$ and $N_{Ga}$ with $\sigma = 1$ are presented using black and red colors. The $N_{Ga}^{1.005}$ presented by blue and the $N_{Ga}^{0.995}$ presented by purple are 0 state $N_{Ga}$ with $\sigma = 100.5\%$ and $\sigma = 99.5\%$.

### 3.2. Transition Channel and SPE Quality of Single-Atom Defects in GaN

Figure 3a illustrates the band structure of $Ga_N^0$ and $N_{Ga}^0$ with a strain of $\sigma = 1$. For $Ga_N^0$, it is clear that the transition energy from $a_2$ to $a_3$ in the spin-down channel (0.5 eV) and the spin-up channel (0.6 eV) are both too small. Given that the zero-phonon line (ZPL) between those defect levels is typically even lower than the transition energy (usually half or less), the luminescence wavelength of $Ga_N^0$ may exceed 4000 nm, which is far from telecommunication windows. On the other hand, for $N_{Ga}$, the highest occupied state $a_1$ is isolated with a VBM of 0.3 eV, which avoids interference between defect levels and electronic states of the host material. The transition energy between $a_1$ and $a_2$ is 3 eV, which is large enough to avoid the influence of thermal excitations. These findings suggest that the defect levels of $N_{Ga}^0$ could form a two-level system that satisfies SPE requirements. However, under additional strain, the defect level of $N_{Ga}$ would be modified. The green and yellow arrows in Figure 3a represent the movement of defect levels when compressive and tensile stress is applied. When compressive strain is added, $a_1$ and $a_2$ ($a_3$) move to a higher energy location, and the transition energy between $a_1$ and $a_2$ ($a_3$) increases. Conversely, tensile strain causes the transition energy to decrease, and those defect levels move in the opposite direction. The specific numerical changes are shown in Table 1. Although defect levels $a_1$ and $a_3$ ($a_2$) change by 0.2 eV as the strain changes from 100.5% to 99.5%, the transition energy between $a_1$ and $a_2$ ($a_3$) remains almost unchanged (less than 0.03 eV). The effect of strain on $N_{Ga}$ will be discussed in more detail in the next section.

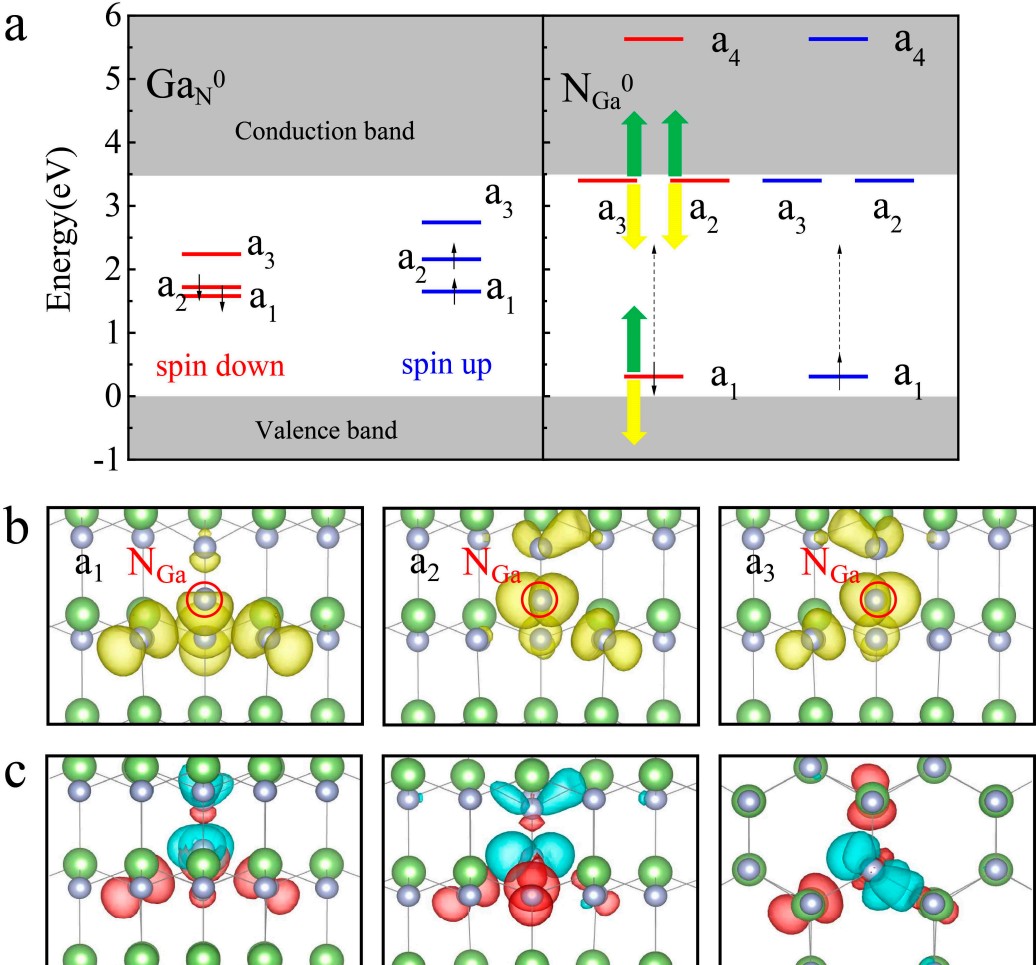

**Figure 3.** (**a**) band structure of $Ga_N$ and $N_{Ga}$ with strain σ = 1. The green and yellow arrow represents the movement of defect levels when the compressive strain and tensile strain are applied. The dashed arrows represent the transition channel. (**b**) the charge densities of $a_1$–$a_3$ of $N_{Ga}$. (**c**) The front view, the side view, and the top view of the charge density difference are calculated by subtracting the charge of $a_2$ from $a_1$. The red and blue colors represent positive and negative values.

**Table 1.** Location of defect levels of $N_{Ga}$ under different strains.

| strain σ | 99.5% | 100% | 100.5% |
|---|---|---|---|
| $a_1$ (eV) | 0.459 | 0.334 | 0.224 |
| $a_2$ (eV) | 3.537 | 3.400 | 3.280 |
| $a_2$-$a_1$ (eV) | 3.078 | 3.066 | 3.056 |

The charge densities of $a_1$-$a_3$ of $N_{Ga}$ at the gamma point are depicted in Figure 3b. These densities are mainly localized around the nitrogen substitution atom rather than being spread throughout the crystal. This suggests that the defect levels exhibit weak interaction with the host, which is advantageous for enhancing the purity of an SPE. Furthermore, the symmetry of the charge densities of $a_2$ and $a_3$ are identical, indicating that these two defect levels is degenerate at the gamma point. Figure 3c displays the charge density difference obtained by subtracting the charge of $a_2$ from that of $a_1$. The red and blue colors in the figure represent positive and negative values, respectively. As the electron transitions from $a_1$ to $a_2$, the charge density progresses from the red part to the blue part. It is evident that the three nitrogen atoms, which are the second nearest neighbors of $N_{Ga}$, primarily contribute to the red part. In contrast, the $N_{Ga}$ itself and the nitrogen atoms

located at the upper nearest neighbor along the [0001] direction mainly contribute to the blue part. In other words, the charge density transfer during the transition occurs mainly from the second nearest neighbors of $N_{Ga}$ to both the $N_{Ga}$ itself and the nearest neighbor nitrogen. At $a_2$, one of the second nearest neighbor nitrogen atoms provides less charge than the other two due to the non-uniform distribution of charge density among these three atoms.

### 3.3. Defect Structure under Strain

Previous reports have proposed using strain to tune the luminescent properties of the defect single-photon emitter (SPE) in an effective way [19]. To analyze the geometry of the defect structure under strain, we calculated the bond length between atoms near $N_{Ga}$. Figure 4a,b shows the bond lengths between $N_{Ga}$ and the nearest neighbor nitrogen atoms along the [0001] direction, as well as between the $N_{Ga}$ and the second nearest neighbor nitrogen atoms. Table 2 summarizes the numerical changes in these bond lengths. Under triaxial compressive strain, $d_1$ (the bond length between $N_{Ga}$ and the upper nearest neighbor nitrogen atom) increases, while $d_2$–$d_4$ and $d_5$–$d_6$ (the bond lengths between $N_{Ga}$ and the second nearest neighbor nitrogen atoms, and between the nearest neighbor nitrogen atoms and the second nearest neighbor nitrogen atoms) decrease. The opposite trend occurs under tensile strain. As discussed earlier, we know that the transition progress is realized primarily by transferring charge density from three second nearest neighbor nitrogen atoms to the $N_{Ga}$ itself and the upper nearest neighbor nitrogen. The decrease in $d_2$–$d_4$ and $d_5$–$d_6$ under compressive strain promotes this process, leading to a lower ZPL. Conversely, the increase in these bonds under tensile strain leads to a higher ZPL. Figure 4c–e shows the atom structure of $N_{Ga}$ in the ground and excited states. We observe that the $N_{Ga}$ in the excited state deviates noticeably from its original central position between the second nearest neighbor nitrogen atoms in the ground state. This deviation can be explained by the non-uniform charge transition in these three atoms from $a_1$ to $a_2$.

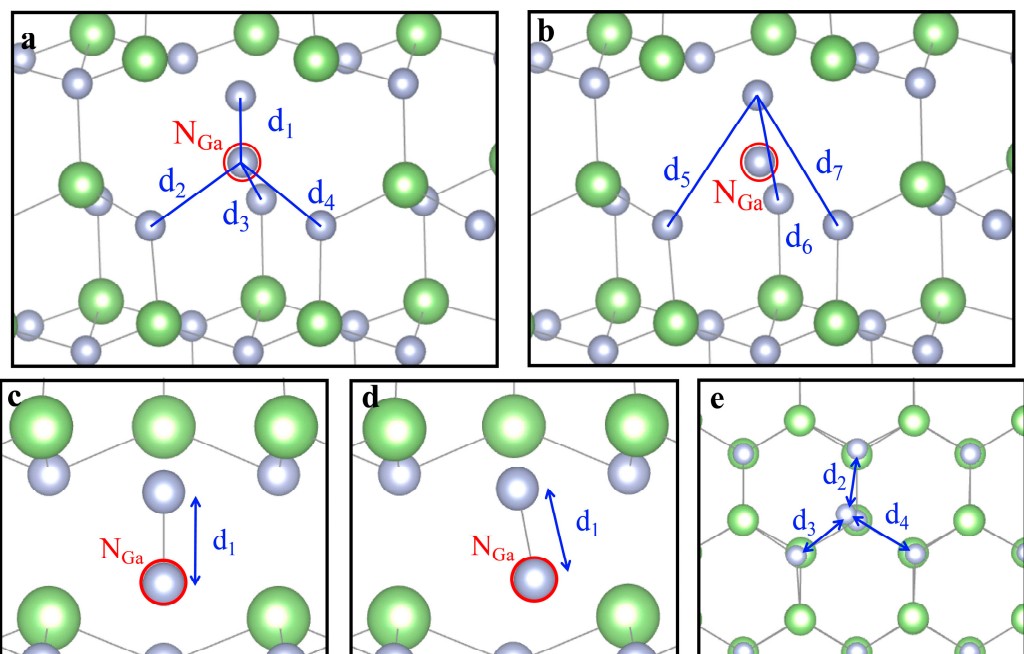

**Figure 4.** Structure diagram of $N_{Ga}$ and bond length between atoms near $N_{Ga}$. (**a,b**) are the bond lengths the bond length between atoms near $N_{Ga}$. (**c,d**) are the side view of $N_{Ga}$ in a ground state and excited state. (**e**) is the top view of $N_{Ga}$ in an excited state.

**Table 2.** The bond lengths of $N_{Ga}$ under different strains.

| State | Ground State | | | Excited State | | |
|---|---|---|---|---|---|---|
| strain σ | 99.5% | 100% | 100.5% | 99.5% | 100% | 100.5% |
| $d_1$ (Å) | 1.30316 | 1.28135 | 1.26637 | 1.31961 | 1.29306 | 1.27982 |
| $d_2$ (Å) | 2.97155 | 3.01326 | 3.05002 | 2.94662 | 3.02600 | 3.07573 |
| $d_3$ (Å) | 2.97246 | 3.01415 | 3.04992 | 3.03867 | 3.06722 | 3.10357 |
| $d_4$ (Å) | 2.97159 | 3.01335 | 3.05011 | 2.94824 | 3.02790 | 3.07888 |
| $d_5$ (Å) | 2.13592 | 2.18277 | 2.22577 | 2.02450 | 2.10169 | 2.15069 |
| $d_6$ (Å) | 2.13668 | 2.18351 | 2.22578 | 2.38010 | 2.45753 | 2.52477 |
| $d_7$ (Å) | 2.13595 | 2.18284 | 2.22584 | 2.02631 | 2.10348 | 2.15338 |

*3.4. The Single Photon Emission Property under Strain*

To evaluate the SPE performance of $N_{Ga}$, we calculated its ZPL and lifetime. Figure 5 shows the configuration coordinate diagram of the optical transition, where state 1 represents the ground state with an electron configuration of $a_1{}^2a_2{}^0$, and state 2 represents the excited state formed by exciting an electron from $a_1$ to $a_2$. The photo absorption (PA) process involves an electron transition from the ground state to the excited state with the same atom configuration. ZPL is the energy difference between the lowest energy of states 1 and 2. For $N_{Ga}$ with a value of σ equal to 1, we calculated the PA and ZPL energies to be 1.01 eV ($E_{PA}$) and 0.86 eV ($E_{ZPL}$), respectively. The calculated PA energy of 1.01 eV suggests that the SPE can be excited by a near-infrared laser, such as the commercial 1064 nm laser. It is worth noting that the calculated ZPL energy of 0.86 eV (1442 nm) falls perfectly within the telecommunication windows.

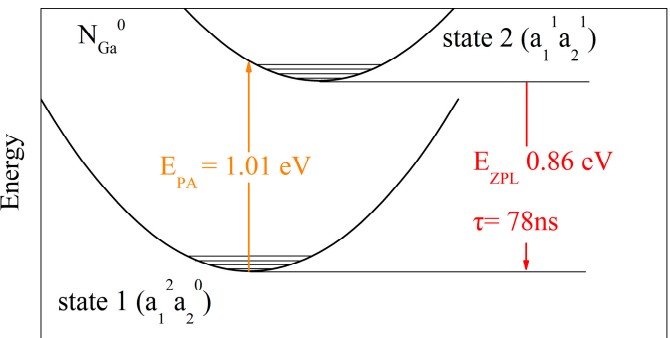

**Figure 5.** Configuration coordinate diagram of the transition process between state 1 and state 2 in $N_{Ga}$ without strain.

The radioactive lifetime τ associated with ZPL can judge the quality of a SPE, which is given by $\tau = \frac{1}{r_{ij}}$. The transition rate $r_{ij}$ can be given by Fermi's golden rule as:

$$r_{ij} = \frac{\omega^3 n |\mu_{ij}|^2}{3\pi\varepsilon_0 \hbar c^3} \tag{2}$$

where $\hbar$ is the reduced Planck constant, $\omega$ is the frequency of the emitted photon of ZPL, $|\mu_{ij}|$ is the transition dipole moment from state $i$ to state $j$, $n$ is the refractive index, $\varepsilon_0$ is the vacuum permittivity, and $c$ is the speed of light in vacuum, respectively. The transition channel in $N_{Ga}$ could provide a two-level system. For $N_{Ga}$ without strain, the calculated lifetime is 78 ns.

Table 3 shows that both the lifetime τ and ZPL are affected when strain is applied. Specifically, the ZPL and lifetime exhibit a near-linear change with triaxial strain; the lifetime increases with triaxial compressive strain, while ZPL decreases with an increase in triaxial compressive stress. The varying strain can modulate the ZPL from 0.849 eV (1460.5 nm)

to 0.984 eV (1260 nm), covering the telecommunication windows from the O band to the E band. Moreover, the lifetime can be modulated from 86 ns to 18 ns. The reduction of ZPL with tensile strain is consistent with the previous discussions regarding charge density and bond. These results suggest that $N_{Ga}$ under strain could meet the requirements for data transmission over optical fibers, and the lifetime could also be reduced to achieve an SPE with higher emission speed. According to Table 3, strain plays a more pronounced influence on lifetime than ZPL. When the triaxial strain changes from 99.5% to 100.5%, the lifetime changes about 4.8 times (from 18 ns to 86 ns). This significant change in the lifetime is large enough for experimental observation. However, the experimental verification still requires more effort due to challenges in the fabrication and identification of this $N_{Ga}$ defect in GaN crystal.

**Table 3.** The lifetime and ZPL of $N_{Ga}$ under different strains.

| Strain | 99.5% | 100% | 100.5% |
|---|---|---|---|
| Lifetime $\tau$ (ns) | 86.002 | 78.928 | 18.028 |
| ZPL (eV) | 0.84921 | 0.86482 | 0.98468 |

## 4. Discussion

Although defects-based RT SPEs in III-nitrides have been reported and gained increasing attention in recent years, their underlying mechanisms remain to be further investigated. This work focuses on the strain-induced tunability of RT SPEs in the telecom band, and the calculations reveal the relationship between the strain (i.e., lattice parameters). luminescence lifetime and ZPL, which provides guidance on the strain conditions required to produce SPE. In addition, our calculations also indicate that changing the strain may work as a novel way to tune the properties of the defects in III-nitrides. When compared with changing temperature or voltage, strain regulation of solid crystals can provide a broader adjusting range in emission wavelength, which owns the enormous potential to achieve RT SPE in GaN working in the telecommunication band. However, the anisotropy of GaN can complicate the effect of stress on the structure. In our study, we only investigate the influence of the triaxial strain uniformly applied along the x-axis, y-axis, and z-axis directions, where the triaxial change is in equal proprtion. In future research, un-uniform strain needs to be further considered, such as uniaxial or biaxial strains along a particular crystal axis or a particular crystal plane.

## 5. Conclusions

We conducted a systematic investigation of the single photon emission properties of $N_{Ga}$ in GaN under different triaxial strains using first-principles calculations. Our results demonstrate that $N_{Ga}$ can serve as a bright and room-temperature single photon emitter. In particular, the addition of a triaxial strain can modulate the ZPL of $N_{Ga}$ from 0.849 eV to 0.984 eV, covering the telecommunication windows from the O band to the E band, and the lifetime $\tau$ from 18 ns to 86 ns. The calculated charge densities and bond lengths suggest that the transition process is mainly affected by the distance from the $N_{Ga}$ defect to its three second-nearest neighbor nitrogen atoms and its upper nearest-neighbor nitrogen. Our work sheds light on the possible origin of defect-based single photon emission in GaN and provides an effective method to achieve room-temperature single photon emission in the telecommunication windows.

**Author Contributions:** Conceptualization, Q.L. and F.C.; methodology, J.Y.; software, J.Y. and K.W.; validation, Q.L. and F.C.; formal analysis, Q.L. and F.C.; investigation, Q.L. and F.C.; resources, Q.L.; data curation, J.Y.; writing—original draft preparation, J.Y.; writing—review and editing, J.Y. and K.W.; visualization, J.Y.; supervision, Y.H. and F.C.; project administration, Q.L. and F.C.; funding acquisition, K.W. and Q.L. All authors have read and agreed to the published version of the manuscript.

**Funding:** This research was funded by [National Natural Science Foundation of China] grant number [61704162], [11604227], [61704163], [11804239], [NSAF U1830109], [State Key Laboratory of Infrared Physics Open Project] grant number [M201917], [Science Challenging Project] grant number [TZ2016003].

**Institutional Review Board Statement:** Not applicable.

**Informed Consent Statement:** Not applicable.

**Data Availability Statement:** Data are contained within the article.

**Conflicts of Interest:** The authors declare no conflict of interest.

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
