# Peer review of "First-Principle Prediction of Stress-Tunable Single-Photon Emitters at Telecommunication Band from Point Defects in GaN"

_photonics, doi:10.3390/photonics10050544_

Round 1

Reviewer 1 Report

The authors reported theoretical analysis on the intrinsic atom defects in GaN through first principle calculation. The work is interesting with some novelty, and could be published after addressing the following issue. Authors should provide references to other SPE systems: quantum dots, transition metal dichalcogneides etc. For example, Nanomaterials, 11, 4, 916 (2021), and ACS Photonics 8, 1069-1076 (2021).

Author Response

Thanks for the comments, we have cited those references in the up-date manuscript.

Reviewer 2 Report

This is a theoretical paper discussing the properties of one type of point defects in GaN based on DFT calculations, in the context of strain-tunable single photon emitters (SPE) at the telecom infrared. Although defects in III-nitrides are rather well known and studied for years, the context of using them for room temperature SPEs is relatively new, and indeed has brought increasing attention recently. The presented results might be of some interest for the community, however, the novelty lies mainly in strain tuning of this particular type of a defect-based SPE (and the active transition lifetimes, which limits the maximal photon generation rate). Many of the other mentioned properties have been reported including strain effects and telecom single photon emission (partly in the previous work of the same Authors - ref. 17, who already indicated the potential of these defects in the telecom range). Independently of that, I would see a space for this short communication in the literature and therefore, I suggest the paper to be accepted for publication in Photonics, after careful manuscript proofreading as there can be found some misspells and mistakes in English. 

Author Response

Thanks for the comment. We have revised the whole manuscript carefully in the up-date manuscript, including correcting the language issues.

Reviewer 3 Report

In this paper, the structure of a telecommunication single-photon emitter found in GaN was inferred by calculating formation energy, band structure, charge densities, structure diagram (bond length) under stress/compress, and so on. According to their results, it is concluded that most probable structure of the telecommunication single-photon emitter was N_Ga because their formation energy was comparable regardless of strain aroused in surrounding crystal and because of their ZPL energies. In addition, they revealed that ZPL energy and its radiation life time clearly corresponds to the surrounding stress, which sounds very interesting results for the corresponding research field.

Although this manuscript in the current form has a sufficient quality for publication, the reviewer recommends the authors to revise it in the minor points or add some explanation for a better readability, as follows:

1)  (P.4, L.130) Please explain how to give the strain (sigma) to the atomic structure model in the simulations. Also, the sigma must be defined in the text when firstly appeared.

2)  (P.4, L.142) “For Ga_N^0, it is clear that the transition energy from to in the … both too small.” How do you obtain the transition energies? In other words, which gaps (a_1 / a_2 / a_3) are the energies corresponding to? Neither Ga_N nor N_Ga has the telecommunication energy range. Besides, the words, “from to” were typo?

3)  (P.6, Table 2) Table 2 has not been referred in the text.

4)  (P.7, L. 232/233) “… provide a two-level system, For N_Ga without strain …”

Comma after “system” should be period?

As mentioned above, the reviewer recommends the authors to give some minor revisions before the publication in this journal.

Author Response

1)  (P.4, L.130) Please explain how to give the strain (sigma) to the atomic structure model in the simulations. Also, the sigma must be defined in the text when firstly appeared.

Answer: Thanks for this comment. The strain σ is described by: σ= a/a0, where a0 is the optimal lattice constant when no strain is present, and a is the corresponding lattice constant when the strains exist. The related description has been corrected in the new manuscript. In the simulation, the strain was given by change the lattice constant along the x-axis, y-axis, and z-axis direction the same time.

2)  (P.4, L.142) “For Ga_N^0, it is clear that the transition energy from to in the … both too small.” How do you obtain the transition energies? In other words, which gaps (a_1 / a_2 / a_3) are the energies corresponding to? Neither Ga_N nor N_Ga has the telecommunication energy range. Besides, the words, “from to” were typo?

Answer: Thanks for this comment. The correct description is “the transition energy from a2 to a3 …’’, and has been revised in the new manuscript. The gap corresponding to transition energy is from a2 to a3 and from a1 to a2 for GaN and NGa, respectively.

  For SPE working in telecommunication energy range, the transition energy should  be greater than telecommunication energy range, because the zero-phonon line (ZPL) is typically even lower than the transition energy (usually half or less). So the 3 eV transition energy for NGa is suit for SPE in telecommunication energy range while the 0.6 eV transition energy for GaN is not。

3)  (P.6, Table 2) Table 2 has not been referred in the text.

 Answer: Thanks for this comment. Table 2 is referred to L.209 P.6, where the sentence is “Table 2 summarizes the numerical changes in these bond lengths. Under triaxial compressive strain …”. We mistook Table 2 as Table 1 in the last vision.

4)  (P.7, L. 232/233) “… provide a two-level system, For N_Ga without strain …”

 Comma after “system” should be period?

Answer: Thanks for this comment. It is period, and has been corrected in the new manuscript.

Reviewer 4 Report

Manuscript ID:     photonics-2324667
Title:         First-principle prediction of stress-tunable single-photon emitters at
            telecommunication band from point defects in GaN
Authors:         Junxiao Yuan, Ke Wang, Feiliang Chen, Qian Li, Yidong Hou

The mauscript studies point defects in GaN employing density functional theory simulations. The authors focus on point defects with excitation energies within telecommunication bands which might act as single photon emitters and could be used in future quantum devices. They show that specific defects exist whose excitation energy is in the right window and which can be tuned / adjusted by applying stress. Furthermore, the authors shed some light on the details of the excitation mechanism.

The manuscript is well written and the results are interesting. I feel, however, that the authors should be more explicit either by comparing their reuslts with experiments or by proposing experiments to test the theoretical predictions. Although qualitatively a very good tool, density functional theory has notoriously problems to quantitatively determine transition energies exactly. In order to pin down their results the authors should point out which experimental results could fix their results quantitatively. Thus, at the moment I hesitate to recommend publication.

Author Response

Thanks for this comment. Our study is mainly to give a susceptibility regulating luminescence lifetime by strain and this conclusion is theoretically self-consistent. Even if the actual value ZPL is slightly different from the calculated result due to the problem of density functional theory, it can be corrected by strain tuning.

As for testing the theoretical predictions in experiment, our theoretical predictions could be verified by measuring luminescence lifetime and adjusting the strain. According to Table 3, strain has a more pronounced effect on lifetime compared to ZPL. When the triaxial strain changes from 99.5% to 100.5%, the lifetime changes about 4.8 times (from 18ns to 86ns). This significant change in lifetime can be easily observed in experiments if the NGa defect can be identified under a microscope. However, fabricating such a two atoms defect pair in a controlled manner and accurately identifying it in a GaN crystal is a challenging task. Therefore, this prediction should be verified in the future when the experimental technology is improved.

Our calculated ZPL could match with experiment result [Y. Zhou; Z. Wang; A. Rasmita; S. Kim. Room temperature solid-state quantum emitters in the telecom range. Sci. Adv. 2018, 4, eaar3580]. The difference of lifetime may caused by temperature, which is 0 K in our relax calculation and room temperature in experiment. The lifetime would decrease with the increase of temperature.

Thank for your comment, this discussion about testing the theoretical predictions has been added in the new manuscript.

Table 3

Strain

99.5%

100%

100.5%

Life time τ (ns)

86.002

78.928

18.028

ZPL (eV)

0.84921

0.86482

0.98468

Compare with experiment

ZPL (eV)

lifetime (eV)

this work (without starin)

0.86

79

this work (with starin)

0.84-0.98

86-18

experiment in Sci. Adv. 2018, 4, eaar3580.

0.89-1.16

0.73

Round 2

Reviewer 4 Report

The authors have adequately adressed my previously raised concerns.